# OpenReview forum: "CPSample: Classifier Protected Sampling for Guarding Training Data During Diffusion"
_NeurIPS.cc/2024/Workshop/SafeGenAi — SafeGenAi Poster_

### Official Review · Reviewer_WPPM · 2024-10-09
**Review of "CPSample: Classifier Protected Sampling for Guarding Training Data During Diffusion"**

**Rating:** 8
**Confidence:** 3

**Review:**

The paper introduces CPSample, an innovative approach to safeguarding training data within diffusion models. Unlike previous methods that may degrade image quality or require retraining of models, CPSample employs a classifier to guide the sampling process away from replication.

**Pros:**

1. **Privacy Protection**: CPSample effectively prevents the replication of training data, providing a layer of privacy protection without significantly compromising image quality.

2. **Computational Efficiency**: Unlike methods that require retraining a diffusion model, CPSample only necessitates the training of a classifier, which is more computationally efficient.

3. **Quality Preservation**: The method achieves state-of-the-art image quality, as evidenced by competitive FID scores on benchmark datasets like CIFAR-10 and CelebA, while preventing exact replication of training images.

**Cons:**

1. **Scalability**: CPSample may face challenges when scaling up to very large datasets, as training a classifier that can generalize well across a vast number of images might be difficult.

2. **Complexity of Training**: Obtaining a classifier with both high accuracy and a stable Lipschitz constant could be complex and may require advanced regularization techniques.

3. **Dependency on Hyperparameters**: The method's effectiveness may rely on the careful tuning of hyperparameters, such as the tolerance threshold and scale parameter, which might require substantial experimental effort to optimize.

The logic structure of the paper is rather clear, and experiments indeed demonstrate the effectiveness of the proposed CPSample. However, as the authors have noted, CPSample employs classifier guidance and does not provide a guarantee against the degradation of image quality.

---

### Official Review · Reviewer_7eov · 2024-10-09
**The paper proposes an efficient and innovative method to safeguard training data in diffusion models through classifier-guided sampling, achieving strong privacy protection and high image quality, though further testing across diverse datasets and deeper comparisons with differential privacy techniques could strengthen the work.**

**Rating:** 6
**Confidence:** 1

**Review:**

This paper presents a novel approach aimed at preventing training data replication in diffusion models without requiring model retraining. By utilizing a classifier trained on random labels to guide the generation process away from training data, CPSample delivers an efficient and scalable solution to privacy concerns, backed by strong empirical results on datasets like CIFAR-10 and CelebA while maintaining minimal impact on image quality. Additionally, the method improves robustness against membership inference attacks, making it highly practical for real-world applications. However, some theoretical limitations exist, particularly concerning the classifier’s Lipschitz constant, and the evaluation would benefit from broader testing on more diverse datasets and a more detailed comparison with differential privacy methods. Despite these areas for improvement, CPSample offers a significant contribution to privacy and efficiency in generative models, with solid theoretical backing and practical applications.

### **Pros**:

- **Novel Approach**: CPSample introduces a unique method of using classifier-guided sampling for data protection, eliminating the need for retraining diffusion models. The classifier, trained on random labels, effectively steers the generation process away from training data, presenting a new perspective on privacy in diffusion models. This approach is simpler and less complex compared to other techniques, while still being highly effective.
- **Strong Empirical Results**: The approach demonstrates substantial success in reducing the similarity between generated images and training data, achieving impressive FID scores and effective results in experiments using CIFAR-10 and CelebA datasets. CPSample successfully prevents data replication while maintaining high image quality, addressing privacy concerns without sacrificing performance.
- **Efficiency**: One of CPSample’s key strengths is its computational efficiency. It circumvents the need for retraining diffusion models by requiring only a one-time training of a classifier, making it far less resource-intensive and more practical for real-world implementations, especially compared to methods that involve retraining for each privacy adjustment.

### **Cons**:

- **Theoretical Weaknesses**: One limitation of the paper is its reliance on assumptions about the classifier's Lipschitz constant, which affects the privacy guarantees. A more rigorous evaluation of this constant could provide stronger theoretical support, but this remains unexplored, weakening the theoretical robustness of the approach.
- **Limited Scope of Evaluation**: The evaluation focuses primarily on CIFAR-10 and CelebA, leaving questions about the method's generalization ability to more complex or sensitive datasets. Additional testing on diverse datasets would provide stronger evidence of its broad applicability.
- **Lack of Detailed Comparison with Differential Privacy**: Although the paper touches on differential privacy, a more comprehensive comparison with state-of-the-art differential privacy techniques would offer a clearer picture of the trade-offs involved. Understanding how CPSample balances privacy and image quality compared to these methods would be valuable for evaluating its relative strengths.

---

### Official Review · Reviewer_p8DP · 2024-10-10
**Review of CPSample**

**Rating:** 7
**Confidence:** 4

**Review:**

This paper presents CPSample as a method to prevent against generation of training data by diffusion models during the inference stage by using a classifier-guided sampling method that steers the generation process away from training data while preserving image quality.

Strengths:

The paper presents a novel approach to guard against training data generation in small datasets where diffusion models are prone to overfitting.

The proposed approach is computationally efficient as it avoids the need for retraining the diffusion model. Instead, it only trains a classifier once, which is significantly less resource-intensive.

The results show that the model balances privacy with fidelity, avoiding a drop in image quality seen in other differential privacy techniques.

Weaknesses:

It would be interesting to see how the method performs on more complex datasets beyond standard benchmarks, such as in fields where privacy is critical (e.g., healthcare), to better understand its effectiveness.

The proposed approach is not strictly differentially private & may still be prone to membership inference attacks.

Although the authors provide a theoretical background to CPSample, the discussion on its formal guarantees of privacy could be more in-depth.

Conclusion:

The paper presents a practical approach for safeguarding training data in diffusion models during inference, using a classifier-guided sampling method that avoids the drawbacks of retraining. The strengths lie in its computational efficiency and effective balance between privacy and image fidelity. However, there is room for improvement in terms of broader evaluation across complex datasets and a deeper theoretical analysis of its privacy guarantees and robustness to membership inference attacks.

---

### Official Review · Reviewer_ZZwm · 2024-10-10
**Adequate Experiments and Thorough Analysis**

**Rating:** 7
**Confidence:** 3

**Review:**

This paper addresses the data privacy concerns associated with diffusion models. The authors introduce a classifier-protected sampling method that preserves the original quality of generated outputs while minimizing the similarity between the training data and the generated data. They establish a theoretical framework and present empirical results across various settings to demonstrate the effectiveness of their method. Although there are some limitations in the empirical and theoretical evidence provided, the results are deemed acceptable. It is recommended that research be conducted to futher enhance and verify the findings.